# OpenReview forum: "Steering Dialogue Dynamics for Robustness against Multi-turn Jailbreaking Attacks"
_TMLR — Accepted by TMLR_

### Review · Reviewer_aeBu · 2025-10-21

**Summary Of Contributions:**

This works proposes a defense to LLM multi-turn adversarial attacks from the perspective of control theory. It models an user-LLM dialogue as a state-space dynamical system, and uses a safety classifier to filter unsafe queries, ensuring the conversation is safety invariant in the face of context drift. Specifically, (i) a neural network $f_\theta(x_{k-1},u_k),g_\theta(x_k, u_k)$ is trained to model current conversation state $x_k$ and predict model response to current query $u_k$, then (ii) a safety predictor $h(x_{k-1}, u_k)$ is trained to quantify safeness of the query. From this, (iii) a Neural Barrier Function checks the safeness of the worse-case adversarial query, to filter out unsafe queries. This provides an safety invariance, that if the current state and the worst-case next step are both within the safe set, then the LLM remains safe in dialogue.

**Audience:**

Yes

**Audience Explanation:**

1. This work addresses an important issue in LLM safety, focusing on multi-turn dialogues where context drift can make jailbreaks and adversarial attacks harder to monitor
2. The proposed method uses concepts from control theory, offering a conceptually novel perspective for designing defenses in LLM systems.

**Broader Impact Concerns:**

No broader impact concerns

**Claims And Evidence:**

Yes

**Claims Explanation:**

Strengths

1. **Novel framing:** Presents an interesting perspective and application of control theory approach to LLM safety and defense.
2. **Clear formulation:** Modeling multi-turn dialogue with state dynamics is intuitive and well-structured; the paper is also clearly written.
3. **Comprehensive experiments:** Includes extensive evaluations of the proposed method, and clearly illustrate the trade-offs between safety, helpfulness, and over-refusal.

Weaknesses

1. **Interdependent system**: The framework involves multiple models (neural dynamics $f$,$g$, and safety predictor $h$). This poses potential challenges since training stability and generalization depend on all components, being sensitive to errors in any one part.
2. **Dependence on training dataset**: Quality of the learned system depends directly on the training dataset to clearly distinguish safe vs. unsafe conversations. Defining "good" dataset and collecting such samples is tricky, and poses uncertainty about generalization to new unseen attacks.
3. **Training hyperparameters**: (a) The joined loss has 4 components; it may be helpful to provide guidance on tuning $\lambda$ coefficients to appropriately balance the loss dynamics. (b) The selection of a suitable steering threshold $\eta$ could be tricky for different attacks to maintain performance and minimize trade-offs.

**Requested Changes:**

My main concerns are listed above, here are some question/minor comment.
1. As stated in Eq(9), one single worst-case query $u_k$ and $u_{k+1}$ among the jailbreaking context set $U$ is used. I could be missing something but how is $U$ defined in practice? Is it a pool of queries computed at each turn on-the-fly? To use the NBF-based filtering, does it require a run of possible attacks to obtain $U$ at each step?
2. It would be helpful to provide some understanding of the training loss dynamics and guidance on balancing each component.
3. It would be helpful to provide examples of over-refusal (filtering out safe queries), to better understand the appropriate range of $\eta$.
4. I am further curious about authors’ view on the role of formal safety guarantees and certificates in the context of LLMs. As such frameworks often appear more theoretical than directly deployable, how do you see these guarantees complementing or differing from more practical, heuristic approaches?

---

> ### Author Response · Authors · 2025-12-12
> **Response to Reviewer aeBu (part 1/4)**
>
> We thank the reviewer for acknowledging our novel framing, clear formulation and comprehensive experiments. We really appreciate the reviewer's constructive suggestions and comments to help improve our work. We have responded to the weaknesses and questions below accordingly. Looking forward to your response and further discussion.
>
>
> > **W1:** Interdependent system: The framework involves multiple models (neural dynamics $f,g$, and safety predictor $h$). This poses potential challenges since training stability and generalization depend on all components, being sensitive to errors in any one part.
>
> Thank you for your feedback. We agree that multiple neural network components, including neural dynamics and safety predictor may be challenging to train, and we conduct further experiments with different weights of training losses in response to **W3(a), Q2** (shown below). Regarding the generalization and sensitivity in any one part, we have conducted experiments in unseen LLM models in Table 1 and Table 2, and unseen attacks in Table 5. The attack success rates are significantly reduced after safety steering with neural dynamics and safety predictor in all the cases, showing the generalizability and robustness of the well-trained neural dynamics and safety predictor on GPT-3.5-turbo over unseen LLM conversations with unseen attacks.
>
>
> > **W2:** Dependence on training dataset: Quality of the learned system depends directly on the training dataset to clearly distinguish safe vs. unsafe conversations. Defining "good" dataset and collecting such samples is tricky, and poses uncertainty about generalization to new unseen attacks.
>
>
> We thank the reviewer for the valuable feedback. We agree that the training dataset will have an influence on the performance of unseen conversations. Therefore,  we conducted experiments with less training data under fewer attack methods and evaluated the performance under unseen attacks. More specifically, we train the safety predictor with fewer attack methods, and compare ASR  of these unseen attack methods and MTBench score with SFT baseline (trained with the same data from fewer attack methods). The results are shown below.
>
> | Train without ActorAttack | llama3-8b-instruct (original) | llama3-8b-instruct+SFT | llama3-8b-instruct+steering |
> |-|-|-|-|
> | ActorAttack ASR | 0.425 | 0.12 | 0.065 |
> | MTBench | 7.96 | 7.43 | 7.67 |
>
> | Train without ActorAttack | Phi4 (original) | Phi4+SFT | Phi4+steering |
> |-|-|-|-|
> | ActorAttack ASR  | 0.405 | 0.085 | 0.065 |
> | MTBench |  8.23 | 7.88 | 7.94 |
>
> | Train without ActorAttack and Opposite-day | llama3-8b-instruct (original) | llama3-8b-instruct+SFT | llama3-8b-instruct+steering |
> |-|-|-|-|
> | ActorAttack ASR | 0.425 | 0.18 | 0.175 |
> | Opposite-day ASR | 0.405 | 0.21 | 0.15 |
> | MTBench | 7.96 | 7.54 | 7.80 |
>
> | Train without ActorAttack and Opposite-day |  Phi4 (original) | Phi4+SFT | Phi4+steering |
> |-|-|-|-|
> | ActorAttack ASR  | 0.405 | 0.16 | 0.155 |
> | Opposite-day ASR  | 0.33 | 0.21 | 0.15 |
> | MTBench | 8.23 | 7.92 | 7.93 |
>
>
> From the tables above, we can see that even though with training data of only 1 or 2 attack methods, our results on unseen attacks are still better than the SFT baseline and our results on MTBench can still beat the baseline as well, showing that our safety predictor can generalize well to unseen attacks and yield a better trade-off of utility compared to SFT method. We have added the experiment results and analysis in Appendix B.2.
>
>  > **Q1:** As stated in Eq(9), one single worst-case query $u \_k$ and $u \_{k+1}$ among the jailbreaking context set $U$ is used. I could be missing something but how is $U$ defined in practice? Is it a pool of queries computed at each turn on-the-fly? To use the NBF-based filtering, does it require a run of possible attacks to obtain $U$ at each step?
>
> Thank you for your questions. The reasonable harmful queries (e.g. jailbreaking context set $U$) empirically follow the context of the attack methods, which have different generation mechanisms and logics for different attack methods (some are pre-generated [4] while others are adaptive according to last response of LLM [5,6]. )
> Since we are focusing on a general attack-agnostic defense, we adopt the query context set $\mathcal{S} \_{context} \^{(k)}$  to represent the union of reasonable queries from different attack methods, which follows policy-independent CBF conditions in control theory. Besides, in practice when using NBF-based filtering, we do not need to explicitly find possible attacks to find $U$ at each turn. Instead, following the Q-filter in the Hamilton-Jacobian reachability [7,8], we directly adopt safety predictor as Q-filter to filter out unsafe queries. The reason why we introduce the worst-case query in Eq (9) is to theoretically bridge the barrier function with the Q-filter (safety filter here).  We have added more clarification in Section 4.3 regarding this in the updated version.

---

> ### Author Response · Authors · 2025-12-12
> **Response to Reviewer aeBu (part 2/4)**
>
> > **W3(a), Q2:** Training hyperparameters: The joined loss has 4 components; it may be helpful to provide guidance on tuning $\lambda$ coefficients to appropriately balance the loss dynamics.  It would be helpful to provide some understanding of the training loss dynamics and guidance on balancing each component.
>
> Thank you for the constructive suggestion. According to your feedback, we have conducted additional experiments and reported the loss dynamics under different $\lambda$ coefficients for safe set loss $\mathcal{L} \_{SS}$, safety invariance loss $\mathcal{L} \_{SI}$, and cross-entropy loss $\mathcal{L} \_{CE}$. When we change each weight, we keep all the other coefficients as default values ($\mathcal{L} \_{SS}=100,\mathcal{L} \_{SI}=100,\mathcal{L} \_{CE}=1$, etc.) We have added Figure 8 to show training dynamics in Appendix B.2 and report the table values below.
>
> As shown in the table below, we can see that when $\lambda \_{SS}$ is too small ($\lambda \_{SS}=10$), $\mathcal{L} \_{SI}$ and $\mathcal{L} \_{SS}$ will remain large along training process, while cross-entropy loss cannot converge to a low value with too large  $\lambda \_{SS}=1000$. Therefore, $\lambda \_{SS}=100$ will be a great balance between all the losses for the training dynamics.
>
> For $\lambda \_{SI}$, it can be seen that $\mathcal{L} \_{SI}$ cannot be reduced to a small number under small weight $\lambda \_{SI}=10$. However, if $\lambda \_{SI}=1000$ is too large, $\mathcal{L} \_{SS}$ and $\mathcal{L} \_{CE}$ will be larger throughout the training dynamics. So the best hyperparameter for weight of safety invariance loss is $\lambda \_{SI}=100$.
>
>
>
> | Different $\lambda_{SS}$ | Training loss | Epoch 0  | Epoch 50 | Epoch 100 | Epoch 150 | Epoch 200 |
> | -  | - | - | - | - | - | - |
> |$\lambda_{SS}=10$| $\mathcal{L}_{SS}$      |   $3.93e^{-3}$   |    $2.47e^{-3}$  |  $1.67e^{-3}$   |   $1.22e^{-3}$  |   $9.78e^{-4}$  |
> |$\lambda_{SS}=100$| $\mathcal{L}_{SS}$     |  $3.75e^{-4}$    |   $1.20e^{-4}$   |  $1.05e^{-4}$   |  $9.19e^{-5}$   |  $8.39e^{-5}$   |
> |$\lambda_{SS}=1000$| $\mathcal{L}_{SS}$   |  $4.55e^{-5}$    |  $4.10e^{-5}$    |  $3.19e^{-5}$   |  $2.90e^{-5}$   |  $2.65e^{-5}$   |
> |$\lambda_{SS}=10$| $\mathcal{L}_{FI}$     |  $3.38e^{-3}$    |  $5.56e^{-5}$    |  $2.43e^{-5}$   |  $1.44e^{-5}$   | $1.04e^{-5}$    |
> |$\lambda_{SS}=100$| $\mathcal{L}_{FI}$    |  $1.04e^{-5}$    |  $6.95e^{-6}$    |  $6.60e^{-6}$   |  $6.11e^{-6}$   | $5.84e^{-6}$     |
> |$\lambda_{SS}=1000$| $\mathcal{L}_{FI}$     |  $3.61e^{-4}$    |   $7.31e^{-6}$   |   $6.68e^{-6}$  |  $6.39e^{-6}$   | $6.04e^{-6}$    |
> |$\lambda_{SS}=10$| $\mathcal{L}_{CE}$      |  1.622    |  1.258    |   1.146  |   1.067  |   1.008  |
> |$\lambda_{SS}=100$| $\mathcal{L}_{CE}$     |   1.600   |  1.265    |  1.162   |   1.096  |   1.050  |
> |$\lambda_{SS}=1000$| $\mathcal{L}_{CE}$       |  1.606    |  1.479    |  1.400   |  1.344   |  1.299   |
>
>
> | Different $\lambda_{SI}$ | Training loss | Epoch 0 | Epoch 50 | Epoch 100 | Epoch 150 | Epoch 200 |
> | -  | - | - | - | - | - | - |
> |$\lambda_{SI}=10$| $\mathcal{L}_{SS}$       |  $1.83e^{-4}$    |      $1.31e^{-4}$  | $1.09e^{-4}$  |  $9.88e^{-5}$  | $9.39e^{-5}$ |
> |$\lambda_{SI}=100$| $\mathcal{L}_{SS}$      |   $3.75e^{-4}$    |   $1.20e^{-4}$   |  $1.05e^{-4}$   |  $9.19e^{-5}$   |  $8.39e^{-5}$   |
> |$\lambda_{SI}=1000$| $\mathcal{L}_{SS}$   |   $2.46e^{-4}$    |     $1.61e^{-4}$  | $1.33e^{-4}$  |  $1.15e^{-4}$  | $1.03e^{-4}$  |
> |$\lambda_{SI}=10$| $\mathcal{L}_{FI}$     | $2.66e^{-5}$      |   $1.89e^{-5}$  |  $1.65e^{-5}$  | $1.59e^{-5}$  |  $1.58e^{-5}$  |
> |$\lambda_{SI}=100$| $\mathcal{L}_{FI}$   | $1.04e^{-5}$    |  $6.95e^{-6}$    |  $6.60e^{-6}$   |  $6.11e^{-6}$   | $5.84e^{-6}$     |
> |$\lambda_{SI}=1000$| $\mathcal{L}_{FI}$     |   $1.59e^{-6}$    |   $7.69e^{-7}$  |  $7.05e^{-7}$  | $6.59e^{-7}$  |  $6.26e^{-7}$   |
> |$\lambda_{SI}=10$| $\mathcal{L}_{CE}$      |   1.601    |   1.265  |  1.176  |  1.111 |  1.060  |
> |$\lambda_{SI}=100$| $\mathcal{L}_{CE}$    |   1.600   |  1.265    |  1.162   |   1.096  |   1.050  |
> |$\lambda_{SI}=1000$| $\mathcal{L}_{CE}$       |  1.598     |  1.320   |  1.227  |  1.160 |  1.106  |

---

> > ### Author Response · Authors · 2025-12-12
> > **Response to Reviewer aeBu (part 3/4)**
> >
> > > **W3(b), Q3:** Training hyperparameters:  The selection of a suitable steering threshold $\eta$ could be tricky for different attacks to maintain performance and minimize trade-offs. It would be helpful to provide examples of over-refusal (filtering out safe queries), to better understand the appropriate range of $\eta$.
> >
> >
> > We thank the reviewer for the constructive suggestions. We agree that suitable steering threshold $\eta$ could be tricky for different attacks to maintain performance and minimize trade-offs. Therefore, we conducted additional experiments over-refusal problems under **more fine-grained and wide-ranging steering thresholds**.
> > More specifically, we systematically evaluate the over-refusal rate on XSTest, JailbreakBench-Benign, and PHTest-Harmless [1] dataset with wide-range steering thresholds ($\eta$) on llama3-8b-instruct and phi4. Regarding the defense baselines, in addition to LoRA-SFT post-training and prompt-based safety steering, we implement two human preference-based post-training mult-turn safety alignment baselines, LoRA DPO [2] and LoRA KTO [3],  and compare them against over-refusal problems.
> >
> > | llama3-8b-instruct                     | XSTest | JailbreakBench-Benign | PHTest-Harmless | ActorAttack ASR |
> > |-|-|-|-|-|
> > | Original                              | 0.078  | 0.34                   | 0.27            | 0.425           |
> > | w/ system prompt                      | 0.178  | 0.49                   | 0.50            | 0.280           |
> > | w/ SFT                                | 0.237  | 0.34                   | 0.44            | 0.070           |
> > | w/ DPO                                | 0.226  | 0.56                   | 0.60            | 0.065           |
> > | w/ KTO                                | 0.100  | 0.40                   | 0.32            | 0.305           |
> > | w/ steering $\eta=1e^{-4}$                  | 0.087  | 0.40                   | 0.31            | 0.075           |
> > | w/ steering $\eta=5e^{-4}$                  | 0.096  | 0.40                   | 0.32            | 0.055           |
> > | w/ steering $\eta=1e^{-3}$                  | 0.096  | 0.41                   | 0.35            | 0.040           |
> > | w/ steering $\eta=5e^{-3}$                  | 0.117  | 0.43                   | 0.38            | 0.025           |
> > | w/ steering $\eta=1e^{-2}$                  | 0.130  | 0.43                   | 0.40            | 0.020           |
> > | w/ steering $\eta=5e^{-2}$                  | 0.243  | 0.47                   | 0.47            | 0.000           |
> >
> >
> > | phi-4                                | XSTest | JailbreakBench-Benign | PHTest-Harmless | ActorAttack ASR |
> > |-|-|-|-|-|
> > | Original                            | 0.100  | 0.18                   | 0.23            | 0.405           |
> > | w/ system prompt                    | 0.052  | 0.17                   | 0.17            | 0.370           |
> > | w/ SFT                              | 0.139  | 0.22                   | 0.24            | 0.100           |
> > | w/ DPO                              | 0.357  | 0.32                   | 0.53            | 0.130           |
> > | w/ KTO                              | 0.117  | 0.22                   | 0.31            | 0.270           |
> > | w/ steering $\eta=1e^{-4}$               | 0.087  | 0.26                   | 0.29            | 0.060           |
> > | w/ steering $\eta=5e^{-4}$               | 0.096  | 0.26                   | 0.32            | 0.030           |
> > | w/ steering $\eta=1e^{-3}$              | 0.100  | 0.27                   | 0.33            | 0.015           |
> > | w/ steering $\eta=5e^{-3}$               | 0.148  | 0.28                   | 0.33            | 0.015           |
> > | w/ steering $\eta=1e^{-2}$               | 0.189  | 0.30                   | 0.44            | 0.010           |
> > | w/ steering $\eta=5e^{-2}$               | 0.337  | 0.36                   | 0.42            | 0.010           |
> >
> >
> >
> > From the tables above, we can see that under three over-refusal datasets, our steering will result in a higher over-refusal rate with larger $\eta$. But compared to the post-training baselines, ours is more flexible and can achieve a better trade-off. With the threshold of $\eta=5e^{-4}$, the over-refusal rate and ActorAttack ASR can be well balanced compared to the baselines, which is consistent with results in Figure 5 and Table 14 regarding practical applicability in the updated version.

---

> > > ### Author Response · Authors · 2025-12-12
> > > **Response to Reviewer aeBu (part 4/4)**
> > >
> > > > **Q4:** I am further curious about authors’ view on the role of formal safety guarantees and certificates in the context of LLMs. As such frameworks often appear more theoretical than directly deployable, how do you see these guarantees complementing or differing from more practical, heuristic approaches?
> > >
> > >
> > > Thank you for the thoughtful question. From my viewpoint, formal safety guarantees and practical heuristic methods are more or less complementary rather than competing. Heuristic approaches are indispensable for real-world deployment because they often work well in practice. At the same time, they are more or less empirical due to black-box neural networks, which need trial and error for better performance.
> > >
> > > Works with formal safety guarantees are usually on the opposite, where people tend to first formulate the problem (e.g. conversation context shifts as a dynamical system) and prove some theoretical results, and then deploy the math framework to the practical problems (e.g. multi-turn defense in our case). The motivation for introducing a control-theoretic perspective is not to claim that we can already offer perfect and deployable guarantees in human-LLM interactions, but to provide a principled way to understand why multi-turn failures occur and how we might prevent them systematically. Formal math tools can help identify the problem from top to bottom and expose structural vulnerabilities that ad-hoc methods might overlook.
> > >
> > > That being said, I fully acknowledge that theoretical frameworks often rely on assumptions that do not hold perfectly in real-world LLM systems. Bridging that gap is exactly what we are working toward, e.g. training *neural* barrier functions and *neural* dynamics to soften those strong assumptions in control theory and bring the benefits of formal guarantees closer to practice. In the long run, I believe theoretical views can help to provide insights for the design of heuristic methods, rather than replace them. Eventually, there might exist some *neuro-symbolic* way that can empirically work well with math interpretability and theoretical insight.
> > >
> > > ---
> > >
> > > [1] An et al. Automatic Pseudo-Harmful Prompt Generation for Evaluating False Refusals in Large Language Models, COLM 2024
> > >
> > > [2] Rafailov et al. Direct Preference Optimization: Your Language Model is Secretly a Reward Model, NeurIPS 2023
> > >
> > > [3] Ethayarajh et al. KTO: Model Alignment as Prospect Theoretic Optimization, ICML 2024
> > >
> > > [4] Ren et al. Derail yourself: Multi-turn llm jailbreak attack through self-discovered clues, 2024
> > >
> > > [5] Russinovich et al. Great, now write an article about that: The crescendo Multi-Turn LLM jailbreak attack. USENIX Security 2025.
> > >
> > > [6] Li et al. Llm defenses are not robust to multi-turn human jailbreaks yet, 2024
> > >
> > > [7] Fisac et al. Bridging hamilton-jacobi safety analysis and reinforcement learning, ICRA 2019
> > >
> > > [8] Li et al. Verifiable Safety Q-Filters via Hamilton-Jacobi Reachability and Multiplicative Q-Networks, L-CSS 2025

---

### Review · Reviewer_M5DU · 2025-10-29

**Summary Of Contributions:**

This paper proposes a safety steering framework rooted in safe control theory, which models multi-turn LLM dialogues via state-space representations and introduces a neural barrier function to proactively detect harmful queries, achieving invariant safety across dialogue turns while outperforming baseline defenses in balancing safety, helpfulness, and over-refusal. It conducts extensive experiments on multiple LLMs (e.g., GPT-3.5-turbo, Claude 3.5 Sonnet) and multi-turn jailbreak attacks (e.g., ActorAttack, Crescendo), validating the framework’s effectiveness and generalizability.

**Audience:**

Yes

**Audience Explanation:**

The research topic addresses a critical need in current LLMs.

**Claims And Evidence:**

Yes

**Claims Explanation:**

- It targets multi-turn jailbreaks instead of only single-turn attacks.
- It uses control theory for dialogue safety, bringing a new technical perspective.
- It works across different LLMs, showing good generalizability not limited to specific models.
- It balances safety, helpfulness, and over-refusal—avoids overly strict filtering that hurts usability.
- It provides lightweight implementations that are easy to deploy.

**Requested Changes:**

- How is the "query context embedding set \(\mathcal{U}_{k-1}\)" exactly constructed? The paper mentions it from attack methods, but lacks details on sampling logic (e.g., how to ensure coverage of "reasonable harmful queries").
- For the state-transition equation \(x_k = f_\theta(x_{k-1}, u_k)\), why choose 3-layer ReLU MLPs with 1536-512-768 dimensions? Is there ablation on network architecture (e.g., layer number, activation function) impact?
- When training \(L_{SI}\), why set \(\kappa=3\) (non-invariant turns) as default? The paper says "empirically best," but lacks comparison with more values (e.g., 1, 5) to justify this choice.
- The safety judge relies on GPT-4o, but does GPT-4o’s scoring bias (e.g., over-labeling certain queries as safe/unsafe) affect NBF training? No analysis on judge reliability is provided.
- The MMLU score drops sharply with \(\eta=1e^{-3}\) (e.g., GPT-4o from 87.04 to 62.69). Why does filtering harm factual knowledge?

---

> ### Author Response · Authors · 2025-12-12
> **Response to Reviewer M5DU (part 1/2)**
>
> We thank the reviewer for the valuable suggestions and constructive feedback. We have responded to the comments below accordingly. Looking forward to your response and further discussion.
>
>
> > **Q1:** How is the "query context embedding set (\mathcal{U}_{k-1})" exactly constructed? The paper mentions it from attack methods, but lacks details on sampling logic (e.g., how to ensure coverage of "reasonable harmful queries").
>
> Thank you for your question. The reasonable harmful queries empirically follow the context of the attack methods, which have different sampling logics for different attack methods (some are pre-generated [1] while others are adaptive according to last response of LLM [2,3]). Since we are focusing on a general attack-agnostic defense, we adopt the query context set $\mathcal{S}_{context}^{(k)}$  to represent the union of reasonable queries from different attack methods, which follows policy-independent CBF conditions in control theory. We would like to refer the audience to these attack papers for more details. We have added more clarification in Section 4.3 regarding this in the updated version.
>
>
>
> > **Q2:** For the state-transition equation (x_k = f_\theta(x_{k-1}, u_k)), why choose 3-layer ReLU MLPs with 1536-512-768 dimensions? Is there ablation on network architecture (e.g., layer number, activation function) impact?
>
> Thank you for your question. To validate the effectiveness of NN-based dynamical system modeling in LLM conversation, we choose the commonly-used implementation of neural dynamics and neural barrier functions from the literature of learning-based control [4,5,6], where small ReLU MLPs are usually easier to verify. The input and output dimensions are chosen to accommodate the size of pre-training language embeddings (all-mpnet-base-v2, all-distilroberta-v1) from Hugging Face. We remark that our framework focuses on validating dynamics and barrier function modeling for LLM conversation, and did not fully explore the potential of different network architecture implementations. Even though there is still room to optimize the network architecture, our current implementation can achieve superior results than the baselines and thus can empirically show the effectiveness of the proposed method.
>
>
>
> > **Q3:** When training (L_{SI}), why set (\kappa=3) (non-invariant turns) as default? The paper says "empirically best," but lacks comparison with more values (e.g., 1, 5) to justify this choice.
>
> Thank you for your suggestion. We have shown the following ablation study of  non-invariant turns $\kappa$ below.
> Since the real non-invariant turns in multi-turn jailbreaking attacks are unknown, empirically $\kappa=3$ mostly results in the best safety steering trade-off, where helpfulness will be higher if $\kappa$ is smaller while steering will likely be stronger with larger $\kappa$. But if  $\kappa$ is infinite, it will be degraded to the case without the safety invariance loss $\mathcal{L}_{SI}$ in Table 6. We have added the results and analysis in Sec. 5.3 in the updated version.
>
> | LLMs                     | $\kappa$ (non-invariant turns) | ActorAttack | Crescendo | Opposite-day | MMLU   | MTBench |
> |-|-|-|-|-|-|-|
> | GPT-3.5-turbo | $\kappa= 2$                   | 0.445       | 0.540     | 0.735        | **67.75** | **8.04** |
> |        GPT-3.5-turbo                  | $\kappa= 4$                     | 0.195       | 0.455     | 0.550        | 63.44  | 7.57    |
> |         GPT-3.5-turbo                 | $\kappa= 3$  (default)          | **0.135**   | **0.430** | 0.655        | 66.24  | 7.93    |
> | Llama-3-8b instruct | $\kappa= 2$            | 0.335       | 0.435     | 0.300        | **65.92** | **8.01** |
> |      Llama-3-8b instruct                    | $\kappa= 4$                     | 0.160       | **0.275** | **0.280**    | 61.65  | 7.50    |
> |      Llama-3-8b instruct                    | $\kappa= 3$  (default)          | **0.120**   | 0.360     | 0.310        | 64.52  | 7.90    |
>
>
> > **Q4:** The safety judge relies on GPT-4o, but does GPT-4o’s scoring bias (e.g., over-labeling certain queries as safe/unsafe) affect NBF training? No analysis on judge reliability is provided.
>
> Thank you for your suggestion. We agree that even though GPT-4o judge is widely used in the literature [7],  it might be biased and we further conducted additional experiments with the judge of GPT-5 under the same prompt as the previous judge. The results have been shown below. We can see that for ActorAttack [1] method, our steering can significantly reduce the attack success rate (ASR) for GPT-3.5-turbo and Llama-3-8b,  which is consistent with the results under judge of GPT-4o.
>
>
>
> | ASR of ActorAttack under Judge GPT-5 | original | with steering |
> | - | - | - |
> | GPT-3.5-turbo     | 0.485     | 0.080     |
> | Llama-3-8b     | 0.350     | 0.070     |

---

> > ### Author Response · Authors · 2025-12-12
> > **Response to Reviewer M5DU (part 2/2)**
> >
> > > **Q5:** The MMLU score drops sharply with (\eta=1e^{-3}) (e.g., GPT-4o from 87.04 to 62.69). Why does filtering harm factual knowledge?
> >
> > Thank you for your question. Since our training is only conducted with the jailbreak trajectories, which are assumed to be the most harmful reasonable queries along the context to ensure invariant safety by barrier function, the model might have out-of-distribution issues with factual knowledge, especially under larger steering thresholds $\eta$.
> >
> > To further deal with concerns about practical applicability due to the trade-off between safety and helpfulness,  we  conduct additional experiments on general utility (e.g. MMLU, MTBench) and ASR with **more fine-grained steering thresholds** $\eta$ below.
> >
> > | llama3-8b-instruct              | MMLU  | MTBench | ActorAttack ASR |
> > |-|-|-|-|
> > | Original                       | 66.00 | 7.96    | 0.425           |
> > | w/ system prompt               | 65.66 | 8.13    | 0.280           |
> > | w/ SFT                         | 63.34 | 7.52    | 0.070           |
> > | w/ steering $\eta=0$              | 63.76 | 7.90    | 0.120           |
> > | w/ steering $\eta=2e^{-4}$           | 62.75 | 7.90    | 0.075           |
> > | w/ steering $\eta=4e^{-4}$          | 60.04 | 7.81    | 0.055           |
> > | w/ steering $\eta=6e^{-4}$           | 55.83 | 7.56    | 0.040           |
> > | w/ steering $\eta=8e^{-4}$           | 51.22 | 7.50    | 0.040           |
> > | w/ steering $\eta=1e^{-3}$           | 46.65 | 7.42    | 0.040           |
> >
> > | phi4                          | MMLU  | MTBench | ActorAttack ASR |
> > |-|-|-|-|
> > | Original                     | 78.49 | 8.23    | 0.405           |
> > | w/ system prompt             | 78.67 | 8.59    | 0.370           |
> > | w/ SFT                       | 76.77 | 8.06    | 0.100           |
> > | w/ steering eta=0            | 76.68 | 8.21    | 0.080           |
> > | w/ steering $\eta=2e^{-4}$         | 74.64 | 8.15    | 0.060           |
> > | w/ steering $\eta=4e^{-4}$         | 71.48 | 8.08    | 0.035           |
> > | w/ steering $\eta=6e^{-4}$         | 66.74 | 7.83    | 0.025           |
> > | w/ steering $\eta=8e^{-4}$         | 61.42 | 7.81    | 0.015           |
> > | w/ steering $\eta=1e^{-3}$         | 56.09 | 7.76    | 0.015           |
> >
> >
> > Interestingly, we can find that with larger $\eta$, MMLU and MTBench scores decay faster while the improvement of ASR gradually saturates compared to the cases with smaller $\eta$. This means that there exists a optimal threshold $\eta \^\*$ between $4e^{-4}$ and $6e^{-4}$ for the best safety-helpfulness trade-off. That being said,  when $0<\eta<\eta\^\*$, the model can become much safer by stronger steering and the helpfulness will not sacrifice too much; but when $\eta>\eta\^\*$, the model's general utility will quickly degrade while the gain in safety tends to saturate and converge if steering continues to be stronger.
> >
> >
> > For practical applicability, combining results of over-refusal in the previous response, we recommend using steering of $\eta=5e^{-4}$ for the best balance of the safety-helpfulness trade-off. Regarding the adaptive mechanism of the steering threshold, we think it is a great idea to explore, and one possible way can be based on inference-time calibration like conformal prediction [8,9]. While it is out of the scope of this work, we believe it is a promising direction for the community as future work. More results and discussion have been added to Appendix B.2.
> >
> > ---
> >
> > [1] Ren et al. Derail yourself: Multi-turn llm jailbreak attack through self-discovered clues, 2024
> >
> > [2] Russinovich et al. Great, now write an article about that: The crescendo Multi-Turn LLM jailbreak attack. USENIX Security 2025.
> >
> > [3] Li et al. Llm defenses are not robust to multi-turn human jailbreaks yet, 2024
> >
> > [4] Zhang et al. Exact verification of relu neural control barrier functions, NeurIPS 2023
> >
> > [5] Hu et al. Real-Time Safe Control of Neural Network Dynamic Models with Sound Approximation, 2024
> >
> > [6] Yang et al. Scalable synthesis of formally verified neural value function for hamilton-jacobi reachability analysis, 2025
> >
> > [7] Qi et al. Finetuning aligned language models compromises safety, even when users do not intend to!, ICLR 2024
> >
> > [8] Cherian et al. Large language model validity via enhanced conformal prediction methods, NeurIPS 2024
> >
> > [9] Chan et al. Conformal Information Pursuit for Interactively Guiding Large Language Models, 2025

---

### Review · Reviewer_BYEM · 2025-11-29

**Summary Of Contributions:**

- The paper treats multi-turn conversations like an evolving system, where each exchange nudges the model's internal state.
- It introduces a neural barrier function that estimates whether the next user query might push the conversation toward unsafe territory.
- Using ideas from control theory, the authors set conditions that keep the dialogue in a safe zone throughout the entire interaction.
- Their approach works as a lightweight add-on that sits in front of any LLM and filters potentially risky queries on the fly.
- In tests across several models and attack methods, this steering method sharply reduces jailbreak success rates while still keeping the models useful.

**Audience:**

Yes

**Audience Explanation:**

- Multi-turn jailbreaks are an active and growing concern in the LLM safety community, so a method specifically designed to defend against them is highly relevant to TMLR readers.
- The paper introduces a novel perspective by framing dialogue safety as a dynamical system problem, which would interest researchers working on both machine learning theory and practical safety.
- The proposed neural barrier function offers a general, model-agnostic approach, making it appealing to practitioners who want deployable safety tools rather than model-specific solutions.

**Broader Impact Concerns:**

There are no specific broader impact concerns.

**Claims And Evidence:**

Yes

**Claims Explanation:**

- The paper provides a clear description of the proposed method and explains each component in a way that is easy to follow.
- It evaluates the approach across multiple LLMs and several established multi-turn jailbreak datasets, which strengthens the reliability of the results.
- The experiments compare the method against diverse baselines, showing consistent improvements in safety while keeping helpfulness intact.
- The authors include analyses on generalization to unseen attacks, demonstrating that the method is not overfitted to a single scenario.

**Requested Changes:**

- The paper misspells “Definition” as “Denifition” in Appendix A.
- The phrase “restated of” is ungrammatical and should be replaced with “restatement of” or “restated version of.”
- In Figures 1, 3, and 4, the meaning of the colors used in the texts or symbols is unclear and should be explicitly described.
- In Equations (2) and (3), $z_k$ is multiply defined, and the notation for $z_k$ in Equation (3) should be changed to avoid ambiguity.
- The phrase “We adopt a NN parameterized” should be corrected to “We adopt an NN parameterized.”
- The authors should define the evaluation metrics mathematically rather than assuming the reader already knows them.
- Table 4 and Figure 5 overlap visually, and their margins should be adjusted to prevent the conflict.
- The LLM models used in this work are somewhat outdated, and more recent models should be used if possible.

---

> ### Author Response · Authors · 2025-12-12
> **Response to Reviewer BYEM**
>
> We thank the reviewer for the valuable suggestions and constructive feedback. We have responded to the comments below accordingly. Looking forward to your response and further discussion.
>
>
> > **Q1:** The paper misspells “Definition” as “Denifition” in Appendix A.
>
> Thank you for pointing this typo out. We have fixed it in the updated version.
>
> > **Q2:** The phrase “restated of” is ungrammatical and should be replaced with “restatement of” or “restated version of.”
>
> Thank you for pointing this out. We have fixed it in the updated version.
>
> > **Q3:** In Figures 1, 3, and 4, the meaning of the colors used in the texts or symbols is unclear and should be explicitly described.
>
> Thank you for pointing this out. We have added clarification in the captions in the updated version.
>
> > **Q4:** In Equations (2) and (3), $z_k$ is multiply defined, and the notation for $z_k$ in Equation (3) should be changed to avoid ambiguity.
>
> Thank you for your suggestion. We would like to clarify that even though there are two formulas regarding $z_k$ in Equations (2) and (3), only (2) is the definition of LLM output embedding $z_k$ while (3) gives the property of observation function $g_\theta$ based on the well-defined LLM output embedding $z_k$.
>
> > **Q5:** The phrase “We adopt a NN parameterized” should be corrected to “We adopt an NN parameterized.”
>
> Thank you for pointing this typo out. We have fixed it in the updated version.
>
> > **Q6:** The authors should define the evaluation metrics mathematically rather than assuming the reader already knows them.
>
> Thank you for your suggestion.  We have added more clarification regarding metrics in the updated version.
>
> > **Q7:** Table 4 and Figure 5 overlap visually, and their margins should be adjusted to prevent the conflict.
>
> Thank you for pointing this typo out. We have fixed it in the updated version.
>
> > **Q8:** The LLM models used in this work are somewhat outdated, and more recent models should be used if possible.
>
> Thank you for your suggestion. We agree that the models are somewhat outdated and we have conducted additional experiments of the most recent GPT-5 and Claude Sonnet 4.5 models, regarding the attack success rate (ASR) and over-refusal rate to show the trade-off. It can be seen that even under the latest powerful models, the original ASR is pretty high while our steering can reduce it under multiple  attack methods, without compromising over-refusal rate too much over the benign data on XSTest dataset. The results further validate the effectiveness and generalizability of the proposed method on the more powerful and capable LLM models. We have added the additional results in Appendix B.2 in the updated version.
>
> | LLMs                     |  ASR (ActorAttack) | ASR (Crescendo) | ASR (Opposite-day) | Over-refusal Rate (XSTest)   |
> |-|-|-|-|-|
> | GPT-5 |  0.355       | 0.350     | 0.325        | 0.052 |
> | GPT-5 + safety steering |  0.040       | 0.150     | 0.105        | 0.122 |
> |        Claude Sonnet 4.5                 |0.540 |  0.370       | 0.250    |    0.035    |
> |        Claude Sonnet 4.5 + safety steering                | 0.110 | 0.300       | 0.135     |   0.074      |

---

### Author Response · Authors · 2025-12-12
**General response**

We would like to express our sincere gratitude to the reviewers and AE for their time and valuable feedback. We have carefully considered all the comments and made a point-to-point response to them. In particular, we have added more clarification and experimental results. All the major changes have been marked in blue in the updated version of our work. We deeply appreciate all the constructive feedback and believe that our changes have  improved the quality of the paper in terms of clear presentation and comprehensive experiments. We are happy to have further discussions if there are any other concerns about our work.

---

### Decision · Action_Editor_sivD · 2026-02-02

**Recommendation:** Accept as is

**Audience:**

Yes

**Audience Explanation:**

Multi-turn jailbreak attacks represent an important and growing challenge in LLM safety. By introducing a control-theoretic perspective and a principled framework for enforcing safety invariance over dialogue turns, this work is likely to be of interest to a broad segment of the TMLR audience, including researchers in machine learning safety, alignment, and trustworthy AI, as well as practitioners concerned with deployable defenses.

**Claims And Evidence:**

Yes

**Claims Explanation:**

The claims of the paper are supported by clear and convincing evidence. The proposed framework is well motivated, the methodology is technically sound, and the experimental evaluation is extensive, covering multiple LLMs, multiple multi-turn jailbreak attacks, and relevant baselines. The results consistently support the paper’s claims regarding improved robustness against multi-turn jailbreaks while maintaining a reasonable trade-off between safety and utility.